# Sensor-Based Prototype of a Smart Assistant for Visually Impaired People—Preliminary Results

**DOI:** 10.3390/s22114271

**Published:** 2022-06-03

**Authors:** Emilia Șipoș, Cosmin Ciuciu, Laura Ivanciu

**Affiliations:** Bases of Electronics Department, Technical University of Cluj-Napoca, 400027 Cluj-Napoca, Romania; ciuciu.co.cosmin@student.utcluj.ro (C.C.); laura.ivanciu@bel.utcluj.ro (L.I.)

**Keywords:** smart assistant, visually impaired, navigation, obstacle detection, route recommendation, intelligent sensors, daily activities

## Abstract

People with visual impairment are the second largest affected category with limited access to assistive products. A complete, portable, and affordable smart assistant for helping visually impaired people to navigate indoors, outdoors, and interact with the environment is presented in this paper. The prototype of the smart assistant consists of a smart cane and a central unit; communication between user and the assistant is carried out through voice messages, making the system suitable for any user, regardless of their IT skills. The assistant is equipped with GPS, electronic compass, Wi-Fi, ultrasonic sensors, an optical sensor, and an RFID reader, to help the user navigate safely. Navigation functionalities work offline, which is especially important in areas where Internet coverage is weak or missing altogether. Physical condition monitoring, medication, shopping, and weather information, facilitate the interaction between the user and the environment, supporting daily activities. The proposed system uses different components for navigation, provides independent navigation systems for indoors and outdoors, both day and night, regardless of weather conditions. Preliminary tests provide encouraging results, indicating that the prototype has the potential to help visually impaired people to achieve a high level of independence in daily activities.

## 1. Introduction

At the end of 2021, an estimated over 2 billion people have near or distance vision impairment. Vision impairment affects people of all ages and poses an enormous global financial burden with the annual global costs of productivity losses. For low- and middle-income regions, the prevalence of distance vision impairment is four times higher than in high-income regions [1]. Vision loss is associated with low income, reduced quality of life, concurrent medical issues, and mental health problems [2]. Visually impaired people are affected not only at a medical level, but also at economical and psychological levels.

The health and well-being of a visually impaired individual and their family can be improved by means of assistive technologies, which also bring broader socioeconomic benefits [3]. Visually impaired people are the second largest affected category with no access to assistive products—more than 200 million people.

Daily struggles of visually impaired people are (and not limited to): lack of ability to identify and interact with the environment, difficulty in outdoor navigation and dependence on others (family/friends) for purchasing groceries and goods.

For a visually impaired person, unassisted traveling requires two levels of navigation: micro-navigation and macro-navigation [4]. Macro-navigation is the ability of the user to know their current location, orientation, and to have information about the route to follow to reach the destination. Micro-navigation involves the user’s ability to identify possible obstacles along the route such as pedestrian crossings, building walls, transparent doors or other obstacles that may interfere with the user.

The classic ways of assisting visually impaired people in outdoor navigation are a white cane, a guide dog, and a human assistant. The white cane is cheap and provides valuable information only about obstacles in front of the user [5]. For safe and correct navigation, the user should know the route perfectly, as well as the position of every crosswalk. A guide dog improves the user’s navigation skills and the ability to handle more difficult situations but this solution is pricey and it can take a while before the request for a trained dog is granted. The third option, having a human assistant by the visually impaired person’s side, is the closest one can get to living a normal life. However, this solution comes with the highest price, and the personal life of the human assistant is also significantly impacted.

New ways of assisting visually impaired people, in both indoor and outdoor navigation, are a direct consequence of the decrease in size of electronic components. Small, non-invasive devices are being prototyped, with varying complexity and functionality. Three main directions are being followed: (1) using sensors to detect obstacles basically in front of the user [6], (2) using cameras for vision-based assistance [7,8], and (3) using radio frequency identification (RFID) [9]. Smart guiding systems are continuously evolving; however, only few of them address both outdoor and indoor navigation [6,7], and their proper and complete use remains low.

### 1.1. Related Work

Paper [10] proposes a system that guides visually impaired people in indoor environments, such as commercial centers, hospitals, and markets. It requires expensive equipment placed in the areas of interest, ultra-wideband sensors, a database to store spatial information about the places, a server, a Wi-Fi network, and a smartphone. A deep neural network is used to identify the user’s location inside a big building. The user activates the application using a vocal command. The application requires an active Internet connection, since it accesses remote resources (cloud).

An augmented white cane is proposed in [11] as part of a prototype of a micro-navigation system that helps the visually impaired to move in indoor environments. The system identifies the position of a person and calculates the velocity and direction of their movements. Using this information, the system determines the user’s trajectory, locates possible obstacles on that route, and offers navigation information to the user. The system consists of an augmented white cane with infrared lights, two infrared cameras, a smartphone and a computer running a software application, which makes the system not portable.

An application was proposed in [12] to help the blind navigate safely in indoor environments such schools, libraries, and shopping malls, based on four main units: navigation, obstacle detection, destination detection and voice command modules. The system depends on computer vision, image processing and pixel manipulation, which are high consumers of computing resources.

Detection of the human skeleton was used to locate people and recommend a safe route for visually impaired pedestrians in [13]. The system consists of RFID cards, an RFID reader, an active Internet connection, a monocular camera, and a smartphone. The user has an RFID reader attached to his ankle; this reader interprets tags positioned under the sidewalk. The location of the user is found by combining the information sent by RFID cards and GPS coordinates. The system was tested in laboratory conditions, with simple scenarios. When the system was used outdoors, where the illuminance changes, the accuracy of object detection and safe route recommendation was reduced.

A system that incorporates a YOLO CNN (You Only Look Once Convolutional Neural Network) to detect, track and recognize, in real time, static and dynamic obstacles encountered during outdoor navigation was proposed in [14]. YOLO recognizes cars, bicycles, and pedestrians, but telephone poles, fences, stairs, and trash cans are only detected as objects. The system requires a new generation smartphone, a laptop equipped with a performant video card, a pair of wireless headphones and Internet connection. As a drawback, the system does not propose an alternative route to avoid identified obstacles.

In [15] a wearable navigation support system for blind and visually impaired people was proposed. The system has four ultrasonic sensors placed at eye level, three on a pair of glasses and one sensor at the right wrist, an Arduino board, and a Beagle Bone Black board. Distances measured using the sensors are the inputs of a logic fuzzy system. The system doesn’t offer localization or other guidance features.

A navigation smart stick based on an ultrasonic sensor, smoke gas sensor, Rx/Tx module, Wi-Fi module, GPS module, Buzzer module, and Text-To-Speech synthesis module was described in [16]. The prototype is rudimentarily built and is based exclusively on the GPS module for outdoor navigation.

The solution proposed in [6] contains sensors to detect objects in proximity and works with pre-recorded messages that help users avoid obstacles. When the device detects an obstacle, the user is informed through an audio message. Unfortunately, with pre-recorded messages, the amount of information transmitted to the user is limited. Recommendation of the fastest and safest route to arrive at a given point in the indoor environment is provided in [17]. The system consists of a smart cane with ultrasonic sensors, camera and accelerometer. The route recommendation depends on the connection with the cloud service.

However, the visually impaired need more than just travel aids for daily activities. Interaction with the environment, such as medication alerts or shopping, is also important.

A mobile application designed to help visually impaired users with their medication is proposed in [18]. The application has five functions: searching for medicinal information, a medication adherence aid and timer, a map directing users to drug stores, a medication history to record individual medication history, and creating the user’s personal medicinal database. After testing, the functions requiring users to input information were proven not to be suitable for visually impaired people. Moreover, voiceover and voice recognition functions were problematic due to pronunciation and the need for IT skills. Users’ ability to use the application depended on their background, IT skills, and experience in using smart technology.

A system designed to assist the visually impaired in shopping activities was proposed in [19]. Guidance in a shopping environment is based on a smart cart, equipped with sensors for detecting obstacles and with an Internet connection. The system offers list creation, information about discounts, shopping history, the direction of exits from the store. The user must create an account in a mobile application, wait for confirmation, and be sure that the store has an available smart cart.

Helping the visually impaired in selecting and matching clothing items is achieved in [20]. Clothes are identified using NFC (Near Field Communication) technology, and their characteristics are uploaded into a web platform. The platform recommends clothes combinations and color palettes.An emergency alert option in systems designed for the visually impaired is found in [21,22,23,24,25,26,27].

The main components used in developing such systems are inertial sensors (gyroscope and accelerometers), GPS, ultrasonic sensors, radar, a standalone camera or smartphone camera, and RFID tags and readers. Usually, these components are attached to an ordinary cane, thus transforming it into an augmented cane. When it comes to features, the most important are indoor and outdoor operation, day and night operation, portability, and emergency alert activation. In addition, route recommendation and condition monitoring are available, in a few systems: route recommendation in [13,17,28,29,30], and condition monitoring in [17,27,31].

While video processing using a camera is often employed in systems that provide reliable day navigation, problems occur in low-light conditions (night). In terms of resource consumption and computational effort, this solution is highly demanding, since processing the video stream must be done in almost real time.

On-chip radar boards can be used in extreme outdoor conditions, providing accurate location measurements. The drawback is the high price of these components, starting from $300.

Inertial and ultrasonic sensors, GPS, and RFID are most commonly employed in developing assistive systems for the visually impaired, due to affordable prices and reasonable location accuracy. The majority of the studied solutions combine GPS and ultrasonic sensors, and only [16] combines GPS, ultrasonic sensors, and RFID. Only a few smart guiding systems address both outdoor and indoor navigation, and their proper and complete use remains low. Moreover, a complex electronic assistant designed to also support visually impaired people in daily activities, together with navigation assistance, has not yet been built, to the best of our knowledge.

A summary of components and features for some representative electronic travel aid systems, including the proposed solution, for the visually impaired is presented in Table 1.

When it comes to pricing, not all papers in Table 1. provide the cost of the proposed system. Based on [30], the minimum price for a cane equipped with sensors is $200, while a complete assistive system can cost as much as $6000.

### 1.2. Motivation

Obstacle avoidance, navigation, and orientation, as well as the design of accessible environments are key elements of modern smart assistants [34]. This paper addresses all three aspects, and presents the prototype of a complete, portable, and affordable smart assistant for helping visually impaired people to navigate indoors and outdoors and interact with the environment.

An ideal assistant for visually impaired people should borrow from the aforementioned three classic solutions (white cane, trained dog, and personal human assistant) and should be able to: help users inside or outside of buildings, guide users in a pedestrian walk, detect all obstacles and inform the user, have non-stop availability, have a positive effect on the user’s self-esteem, reduce dependence on other people, and be affordable.

The goals of the proposed smart electronic assistant are:To help the person in outdoor and indoor environments, daytime and nighttime;To detect obstacles and suggest alternative routes for avoiding them;To provide directions in a new environment (city/building);To assist the person in shopping activities;To send basic medical information to authorized personnel (physician/nurse);To be usable by any visually impaired person, regardless of their IT skills and proficiency;To be available to the user non-stop.

### 1.3. Contributions

The major contributions can be categorized into localization and distance measuring, communication between user and smart assistant, and supporting daily activities.

Localization and distance measuring is achieved by:A sensor-based mechanism for localization, consisting of electronic compass, GPS, ultrasonic sensor, RFID reader and tags;An algorithm for measuring the traveled distance, using a wheel equipped with a high reflection index (trigger) and optical sensor TCRT5000;Separate indoor and outdoor navigation systems;Day and night navigation, regardless of weather conditions;Non-stop availability, even in absence of Internet connection;Communication between user and smart assistant is facilitated by:Easy interaction through voice commands, using Rhasspy Voice Assistant;Voice command recognition without an Internet connection.

Integrating navigation modes and daily activities supporting functions (physical condition, medication, shopping, weather information) in a single system is a key element of improving the interaction between visually impaired people and the environment.

The remainder of the paper is organized as follows: the materials and methods used in implementing the proposed system are described in Section 2. Section 3 shows the results and finally, Section 4 concludes the paper, discusses the advantages and limitations, and points out a series of future developments.

## 2. Materials and Methods

The smart assistant has two hardware parts: the central unit and the smart cane, as presented in Figure 1. The core of the central unit is a Raspberry Pi 3B+ minicomputer, which controls a GPS Adafruit Ultimate module, a gyroscope (electronic compass), an ESP8266-01 module with a serial-to-USB convertor, an audio card (sound processor) with an USB interface, a push button, a microphone, and a speaker. The central unit is supplied by an external battery. The electronic components of the smart cane are a Wemos D1 mini development board, two ultrasonic sensors, an RFID reader, an TCRT5000 reflective optical sensor and a portable battery.

On the other hand, the software components handle the communication between smart assistant and user: voice messages containing commands and instructions are transmitted, received, and interpreted.

### 2.1. Modules of the Central Unit

The electronic part of the central unit is based on a GPS module and an electronic compass, as shown in Figure 2.

#### 2.1.1. GPS Module

A study of the characteristics of the GPS module was conducted, and two GPS modules where selected for further investigation: the Adafruit Ultimate and the NEO-6.

The GPS Adafruit Ultimate module can track up to 22 satellites on 66 channels. Using its internal antenna, it has 1–10 Hz updates and a low current draw, 20 mA. Due to the ultra-low drop-out 3.3 V regulator, it can be supplied with (3.3, 5) VDC. The internal flash memory can store about 16 h of data and provides accuracies of 1.8 m for position and 0.1 m/s for velocity [35]. The Adafruit Ultimate module is used also in [22,23,36], to find the location of blind people.

The GPS NEO-6 module is ideal for battery-operated mobile devices with cost and space constraints. The update rate is 1–5 Hz, it can track up to 22 satellites on 50 channels, and provides accuracies of 2.5 m for horizontal position and 0.1 m/s for velocity. The supply voltage is (2.7, 3.6) V, and activating the power-saving mode results in low power consumption, by turning the module off [37]. The NEO-6 is used in [24,25,26,27] for location monitoring for blind people.

The messages received from the GPS modules are strings of numbers and letters, coded using the MNEA (National Marine Electronics Association) protocol. The information regarding time, latitude, and longitude is extracted from the GPS messages and used for outdoor navigation. Both GPS modules were tested in similar conditions to decide which is suitable for the proposed system. The update frequency rate of the GPS signal is between 1 and 2 Hz. A preset route was chosen to be walked, in the same day, and under the same weather conditions, for both modules. This route was also used in testing the final system for outdoor navigation. Based on the GPS values stored in a database, the map was updated with the values provided by the modules.

Using a sampling algorithm, ten locations were extracted from the database, and through the Google MyMaps app, all ten points were positioned at the exact location. The average deviation between reference GPS values and the values provided by both modules was computed. The measured deviation was 6.7 m for both modules which decreased to 1.5 m after an external antenna was attached to the modules. For the Adafruit module, the maximum measured deviation was 13 m, whereas for the NEO-6 module, the maximum deviation was 14 m. Based on these measurements, the GPS Adafruit Ultimate module was selected. A map with the GPS values obtained from the Adafruit Ultimate module is presented in Figure 3.

#### 2.1.2. Compass Module HMC5883L

The HMC5883L is a triple-axis magnetometer compass module that communicates via the I^2^C interface. The compass module contains an integrated 12-bit analog-to-digital converter and can be supplied with either 3.3 V or 5 V, for use with 3 V or 5 V microcontrollers. The accuracy of the module is 1–2 degrees [38].

The magnetic field value on each of the three axes (*X*, *Y*, *Z*) is stored into an internal 16-bit memory. Data read by the HMC5883L is converted, in order to obtain the difference between the angle defined by the user orientation and the north–south poles direction.

Since the north geographic pole and the north magnetic pole are not the same, the angle for direction used by the smart electronic assistant needs to be calibrated, using the magnetic declination:(1)$$\mathrm{angle}=\mathrm{arctg}\;\left(\frac{Hy}{Hx}\right)+\mathrm{declination},\;\mathrm{if}\mathrm{angle}\;>\;2\times \pi \;\mathrm{then}\;\mathrm{angle}\;=\;\mathrm{angle}\;-\;2\times \pi \;\mathrm{if}\mathrm{angle}\;<\;0\;\mathrm{then}\;\mathrm{angle}\;=\;\mathrm{angle}\;+\;2\times \pi$$
where *Hy* and *Hx* represent the magnetic field values on the *X* and *Y* axes.

To identify the user orientation, only *X* and *Y* axes were used, as seen above. Then, the angle is converted to degrees.

The HMC5883L module was used in [39] to build a cane for the blind, and in [40] to implement a substitute vision system, that uses visible light communication and geomagnetism.

#### 2.1.3. Wi-Fi Module ESP8266

The ESP8266 is a highly integrated Wi-Fi module, with a compact design and reliable performance in the Internet of Things industry; it can perform either as a standalone application or as a slave to a host microcontroller unit. The internal processor of the ESP8266 achieves ultra-low power consumption and reaches a maximum clock speed of 160 MHz. ESP8266 uses external SPI flash to store user programs and supports up to 16 MB memory capacity [41].

The module has a compact design, requires minimal external circuitry, and can be interfaced with external sensors and other devices. Many applications [16,42,43] use the ESP 8266 module in wearable electronics and smart canes for blind people.

### 2.2. Smart Cane

The smart cane consists of an extendable stick, a rubber wheel, and a case to protect the electronic components (Figure 4a). The case was designed and printed with a 3D printer to protect the hardware components from external factors such as dust, mud, slush, etc. The wheel is a durable one and faces all indoor and outdoor challenges. The dimensions of the case are 12 × 12 × 10 cm^3^., and it weighs 285 g, including battery, wheel, and cane. A 1.5-m extendable stick is attached to the case, to safely handle the case and the wheel. Compared to a regular cane, the smart cane brings minimal changes in terms of size and handling, meaning it can be used right away, without additional training.

The electronic components inside the case (Figure 4b) are: (1) Wemos D1 mini development board, (2) two ultrasonic sensors, (4) TCRT5000 reflective optical sensor, (5) RFID reader, and (6) battery.

The circuit diagram, showing the connections between the electronic components in the case, is shown in Figure 4c. The diagram was created in Fritzing, an open-source hardware that fosters a creative ecosystem to document electronic prototypes and lay out professional PCBs [44]. The hardware components are: (1) Wemos D1 mini module; (2) HC-SR04 ultrasonic sensors from the left; (3) HC-SR04 ultrasonic sensors from the right, (4) MRC522 RFID reader; (5) TCRT5000e reflective optical sensor; (6) 5 V portable power bank.

#### 2.2.1. Ultrasonic Sensor HC-SR04

HC-SR04 is a distance sensor with 5 V supply and provides 2 cm to 400 cm of non-contact measurement functionality, with a ranging accuracy of up to 3 mm [45].

Figure 5 illustrates the transmission and reception of signals between the smart cane with its two ultrasonic sensors, and an obstacle detected by the left ultrasonic sensor.

The sensor includes an ultrasonic transmitter, a receiver and a control circuit. The distance (*d*) is directly proportional to the time interval between sending the trigger signal and receiving the echo signal, as computed with Equation (2):(2)$$d=\frac{c\times \left(Ts+Tr\right)}{2}\;$$
where *c* is the speed of sound, *Ts* is the sending time and *Tr* is the receiving time. The signal travels from the smart cane to the obstacle in *Ts* and back in *Tr*, thus double the distance between smart cane and obstacle is travelled.

Being a small and cheap sensor, HC-SR04 is often used in applications for blind people, as in almost all previously mentioned papers.

#### 2.2.2. Reflective Optical Sensor TCRT5000

The TCRT5000 is an infrared sensor, consisting of an infrared emitter and a phototransistor, in a leaded package, with 5 V operating voltage [46]. The sensor emits infrared light and waits for an echo. An object is detected when the echo signal is not received.

To the best of our knowledge, the TCRT5000 has never been used in navigation tools for visually impaired people.

#### 2.2.3. RFID Reader MFRC522

An RFID card must be scanned by an RFID reader, in order to identify the information stored on it. The MFRC522 RFID reader is used for contactless communication, and it contains analog circuitry to demodulate and decode responses. The supply voltage is 2.5 V to 3.3 V. An internal self-test capability and programmable input/output pins are other characteristics for which MFRC522 was chosen as a component of the smart assistant.

The typical operating distance is a maximum 50 mm, depending on the antenna size and tuning. Data transfers up to 10 Mbit/s are achievable and the baud rate of communication is up to 848 kBd. The MFRC522 supports direct interfacing of hosts using serial peripheral interface (SPI), I^2^C-bus, or serial UART interfaces [47].

MRFC522 was used in [12,48] to build indoor navigation systems and in [49] to implement a tactile device for people with communication disabilities.

In indoor conditions, the smart electronic assistant works based on RFID cards placed at different locations on the hallway floor and in front of doors. For this reason, the RFID reader is attached to the bottom part of the smart cane, facing downward.

### 2.3. Communication between User and Smart Assistant

The software part relies on the artificial intelligent Rhasspy Voice Assistant, as well as Python code, and is in charge of the communication between the user and the smart assistant. Rhasspy is an open source, fully offline set of voice assistant services for many human languages [50]. It has a series of a hotkeys for activating the voice recognition mechanism, out of which the Snowboy hotkey is used in the proposed smart assistant. Rhasspy is permanently active in the smart assistant and is independent from the Python code. Rhasspy is the intermediary between the user and the Python code: Rhasspy receives audio signals from the user, converts them to Python commands, then answers back to the user by means of audio signals. Communication between Rhasspy and Python is achieved through the message queueing telemetry transport (MQTT) protocol. MQTT is lightweight and efficient, offers bi-directional communications (between device and cloud), offers reliable message delivery, and uses modern authentication protocols [51].

### 2.4. Measuring the Traveled Distance

Measuring the traveled distance is the way the smart cane keeps track of how much of the proposed route has already been covered. The distance is measured by the TCRT5000 optical sensor, placed on the wheel, which detects the number of wheel spins, based on a high reflection index. The algorithm used for distance measuring is as follows:When the user starts moving, the wheel begins to spin;The IR sensor detects the spinning and the microcontroller starts counting the spins;Data is sent to the central unit, once every 500 ms;The central unit converts the number of spins to distance.

The wheel, equipped with the high reflection index trigger, is shown in Figure 6a. The distance is computed considering the diameter of the wheel and the number of spins detected by the reflective sensor:(3)Distance=π×D×P,
where *D* = diameter of the wheel and *P* = number of the impulses transmitted by the reflective sensor.

The larger the wheel diameter, the greater the distance traveled during a single spin. For example, for a wheel with 8 cm diameter, the distance corresponding to a spin is 25.1 cm.

The measured distance can have an error from 0 cm to the distance corresponding to a full spin, depending on the initial position of the trigger. Both situations are depicted in Figure 6b. The maximum error appears when the trigger passes the IR sensor when the user starts moving. Considering the previous example, the maximum error is 25 cm.

### 2.5. Communication between Electronic Components

To communicate with the GPS and electronic compass modules, the Raspberry Pi minicomputer uses serial and I^2^C ports. Two types of signals are received, converted to a specific format and analyzed by the smart electronic assistant: audio signals and data.

The audio card (Figure 1) receives the signal captured by the microphone, converts it into a digital signal and transmits it to the Raspberry Pi development board. The audio card also performs the backward conversion, from digital to audio signal, sent to the headphones.

Communication between the central unit and the smart cane is based on a server-client model. The ESP8266 is used as an access point and the Wemos D1 board as a slave. When the Wemos D1 transmits data collected through the smart cane, the Wi-Fi module converts the signal and saves the data. The central unit is supplied from an external portable power source.

### 2.6. Navigation

In order to guide a visually impaired person from point A to point B, two levels of navigation must be available: macro-navigation, to compute the shortest distance, and micro-navigation, to detect and avoid obstacles. For the proposed smart assistant, these tasks are achieved by two working modes, *Navigation mode* and *Obstacle mode.* While *Navigation mode* computes the shortest path between current location and destination, and guides the user along the way, *Obstacle mode* identifies obstacles that are in the way and prevents the user from bumping into objects (e.g., furniture) or other people, thus ensuring a completely safe journey. Both modes use vocal commands to inform and guide the user.

#### 2.6.1. Macro-Navigation

The macro-navigation criterion is fulfilled by using Dijkstra’s algorithm, in *Navigation mode*. The user can travel completely safely along various checkpoints placed on a customized map. The map is customized online using Google My Maps, according to the user’s needs. Checkpoints, or markers, represent major locations, such as public institutions, hospitals, train and public transport stations, restaurants, shopping centers, and other intermediate locations considered important for the user. In addition to the markers in these locations, crossroads from which more than one route can be followed must also be marked.

The smart assistant takes into account blind peoples’ needs when the customized map is created. Map customization means setting the checkpoints in Google My Maps and defining paths between checkpoints that are accessible to the visually impaired. Thus, when the shortest path between the current location and the destination is computed, Dijkstra’s algorithm works only with edges that represent accessible paths. The current location is first identified by the GPS module; then, the smart assistant verifies which checkpoint from the database is the closest to the current location. The shortest path between current location and nearest checkpoint is computed using Dijkstra’s algorithm.

Dijkstra’s algorithm finds the shortest path between two nodes in a graph. A comparison between Dijkstra’s algorithm and four other popular pathfinding algorithms, in the IoT context of smart buildings for visually impaired people, can be found in [52]. The study in [52] was conducted only in simulation and only for indoor environments. Dijkstra’s algorithm is also used in [28,53] in a guidance system.

In *Navigation mode*, the nodes in Dijkstra’s algorithm are the GPS locations (the previously set checkpoints) from the database. The paths that connect all nodes are biased with the physical distance between GPS points. The output from Dijkstra’s algorithm provides a vector with nodes in the order to be traveled to reach the desired destination, through the shortest path between the current and desired locations.

The distance between two points is computed using equation (4), as follows:(4)$$\Delta \varphi =\varphi_{2}-\varphi_{1}\;\Delta \gamma =\beta_{2}-\beta_{1}\;a=\sin{\left(\frac{\Delta \varphi }{2}\right)}^{2}+\cos(\varphi_{1})\times \cos\left(\varphi_{2}\right)\times \sin{\left(\frac{\Delta \gamma }{2}\right)}^{2}\;distance=2\times R\times \mathrm{arctg}(\frac{\sqrt{a}}{\sqrt{1-a}})$$
where$\varphi_{1}\;$ is the latitude of the current location, expressed in radians;$\varphi_{2}\;$ is the latitude of the next point, expressed in radians;$\beta_{1}\;$ is the longitude of the current location, expressed in radians;$\beta_{2}$ is the longitude of the next point location, expressed in radians;*R* = 6371 km is the radius of the Earth.

If the user has to change directions (rotate) in order to get to the next intermediate point, the rotation angle, expressed in degrees, is computed using Equation (5)
(5)$$Y=\cos\left(\Delta \gamma \right)+\cos(\varphi_{2})\;X=\cos(\varphi_{1})\times \cos(\varphi_{2})-\sin(\varphi_{1})\times \cos(\varphi_{2})\times \cos(\Delta \gamma )\;Radius=\mathrm{arctg}(\frac{X}{Y})\;degree=\frac{Radius\times 180}{\pi }$$
where Δγ is the difference between $\beta_{2}\;$ and $\beta_{1}$.

The smart assistant, set to *Navigation mode,* computes the route to be followed and the distance to be walked, so as to arrive at the nearest point on the map, and further on, to the final destination. The user is guided by means of vocal commands, containing both direction and angle instructions, such as: “Go forward 45 m”, “Rotate right with 45 degrees”, “Rotate left with 30 degrees”.

The decision to rotate left or right depends on the angle, as shown in Figure 7. Three scenarios are considered: angle negative, angle positive but less than 180 degrees, and positive and more than 180 degrees (reflex angle). The smart assistant transmits the vocal messages to the user.

At every checkpoint on the map (GPS location from the database), the smart assistant informs the user about the new route to be followed (direction and angle). While traveling, the location of the user is permanently monitored; if the deviation from the map is greater than 10 degrees, the user is informed. A vocal message notifies the user about upcoming crosswalks.

#### 2.6.2. Micro-Navigation

The *Obstacle mode* of the smart assistant is designed to fulfill the micro-navigation criterion, that is to identify and inform the user about obstacles in their way. In outdoor navigation, the smart assistant detects obstacles such as borders, benches, other persons, lighting poles, or facades of buildings, and informs the user when they should change direction and/or angle. In indoor navigation, obstacles are pieces of furniture, walls, other persons, stairs, doors, and other objects inside buildings.

The two ultrasonic sensors placed on the smart cane are responsible for obstacle detection. For both sensors, the central axis is rotated 45 degrees, due to the fact that when people are moving in one direction and encounter an obstacle in their path, they tend to turn left or right, at angles from 40 to 50 degrees.

The smart assistant in *Obstacle mode* informs the user if the distance to the detected obstacle is less than a threshold, set to 40 cm, and tells them what to do, as shown in Figure 8. This particular value was chosen based on two considerations: (1) minimum width of a typical hallway (1 m), (2) safe reaction time after obstacle detection. For values above 40 cm, the sensors continuously detect the walls as obstacles, on both sides of the cane, directing the user from one wall to the other. For values below 40 cm, there isn’t enough time for the user to react.

## 3. Results

The prototype of the smart assistant contains the smart cane, the central unit, and headphones and microphone, as shown in Figure 9.

The functionalities of the proposed smart assistant are *Navigation mode, Inside mode, Obstacle mode, Emergency alert, Weather info, Market list, Physical condition, Date and time, Medication schedule* and *Bus alert*. All functionalities are activated/inactivated by the user through vocal commands/messages (Table 2), which are recognized and interpreted by Rhasspy.

The accuracy of correctly identifying different vocal commands was tested in several scenarios (Table 3), for a normal male voice (50 to 60 dB), where “1” represents correct identification and “0” represents incorrect identification of the vocal command. Tests were performed indoors, with gradually increasing background noise by adjusting the volume of the noise source (repetitive low and high frequency sounds). The chosen levels were between 20 dB and 80 dB, with five tests for each level. Up to 50 dB noise, all commands were correctly identified. Once the noise increased above 50 dB, the environment was considered noisy, and the accuracy gradually dropped to 60%, for more than 75 dB noise.

### 3.1. Navigation Mode

In *Navigation mode*, the smart assistant guides the user from the current location to the destination, ensuring user safety throughout the journey. *Navigation mode* is designed to work offline, without the need for an Internet connection, which is especially important in areas where Internet coverage is weak or missing altogether.

Prior to using *Navigation mode*, the database containing the checkpoints is created using the Google My Maps application. A satellite-based map was used to insert markers as checkpoints. Every marker has a specific ID, but a name and a description can also be added. The name can be close to the actual name/purpose of the location, but not necessarily, since *Navigation mode* uses the IDs provided by the satellite maps. After all markers are placed, the entire structure is exported as an *.xml* file that contains names, descriptions and GPS coordinates of all markers. Next, a *.csv* file is saved, containing information about the markers organized as “ID, Latitude, Longitude, Description”. The markers represent the nodes used in Dijkstra’s algorithm and the edges are the lines between markers.

Dijkstra’s algorithm works only with edges that represent accessible paths, the red lines in Figure 10a.

The user can choose a destination from a predefined list. To activate a destination, the user voices a command, in the form of:“Set destination to …” followed by the desired destination, or“I want to go to …” followed by the desired destination.

*Navigation mode* was tested for various locations. In Figure 10b, location 1 is the initial location of the user, and location 4 represents the nearest bus station. To activate *Navigation mode*, the user states the intention: “Set destination to bus station“. *Navigation mode* identifies the current location, computes the shortest path to be followed by the user and determines the intermediate points between the current location and the destination.

First, the distance between point 1 and point 2 and the orientation of the user are computed; the user is informed about the direction and the distance (in meters) from current location to point 2. When the user arrives at point 2, *Navigation mode* computes the rotation angle (denoted A in Figure 10) and the distance to point 3; this information is communicated to the user as “Rotate right with 45 degrees” and “Go forward 45 m”. In point 3, *Navigation mode* updates the user’s location and communicates the new rotation angle (angle B) and the distance to the bus station. The messages transmitted to the user is “Rotate left with 30 degrees” and “Go forward 10 m”. When the user finally gets to the bus station, the message is “Congratulations, you reached your destination!”.

### 3.2. Obstacle Mode

To activate/deactivate *Obstacle mode*, the user says “Obstacle mode enable” or “Obstacle mode disable”.

In *Obstacle mode*, the smart cane and the central unit communicate by means of a wireless connection. Data is transmitted with a 50 Hz frequency Wemos development board, located at the tip of the smart cane, and the ESP8266-01 module, located in the central unit.

The information sent by the smart cane is organized as: “*distanceLeft*, *DistanceRight, StepCounter*, *RFIDtext*”. *DistanceLeft* and *DistanceRight* are the distances measured by the two ultrasonic sensors, *StepCounter* shows the number of wheel spins and *RFIDtext* is a possible text reading from RFID cards, if the smart cane detects any. The threshold for obstacles is 40 cm. The reason for choosing this value is that the necessary time to measure the distance and transmit an audio message to the user is approximatively 1 s, meanwhile, the user approaches at a distance of 20 cm from the obstacle.

Considering that the distance between the user and the smart cane wheel is approximatively 1.5 m, the smart assistant detects obstacles at 1.5–2 m away from the user.

The *Obstacle mode* was tested in a scenario where the user needs guidance to follow a predetermined pathway, without colliding into walls, objects in the way, or other people. The path is shown in Figure 11, and it contains both straight and winding sections. Several tests were carried out, using different speeds, between 0.4 and 1 m/s. The proposed assistant is able to accurately detect obstacles up to a speed of 0.8 m/s, while for other smart navigation aids the speed is lower—0.5 m/s [30] and 0.65 m/s [10].

### 3.3. Inside Mode

Safe navigation is needed whether indoors or out. While outdoor navigation relies on the GPS module, for indoor spaces, the module is usually rendered useless. To help the user navigate safely while indoors, the *Inside mode* was developed. *Inside mode* offers information about the check points placed inside buildings, commercial centers or public institutions, and is based on two components: tags and tag reader. The tags are RFID devices that can be placed on or under the floor. RFID tags store information about the direction to be followed for accessing different rooms, stairs, shops, food areas, etc. To read the information from the tags, an RFID reader was placed at the base of the cane.

The *Inside mode* was tested under real conditions, in the scenario from Figure 12a. The red square represents the RFID tags placed under the floor in the lobby at the entrance of a building. All RFID cards are oriented towards the north pole. When the user is oriented towards the food court, the smart cane scans RFID tag and the message “Food court is forward, Pharmacy is to the right, Market is to the left” is transmitted to the user.

The format of the data read from the RFID tag is “Food_area_nn/Market_ee/Pharmacy_ww/#”. The values “nn”, “ee”, and “ww” represent the orientation, with respect to the north pole, as in N, S, E, W, NE, NW, SE, or SW. When the user scans an RFID tag, *Inside mode* compares the user orientation with the location’s orientation and updates the user in real time. To transmit the cardinal point to the user, directions as “Front”, “Left”, “Right”, “Backward”, “Front-left”, “Front-right”, “Backward-right”, and “Backward-left” are used. Every direction has a range of approximately 45 degrees.

An accurate reading of RFID cards can only be achieved if the distance between tag and tag reader is relatively small. The RFID reader was placed on the tip, facing downward (Figure 12b). The distance between tag and reader was increased from 20 mm to 36 mm, which resulted in a dramatic decrease in accuracy (Table 4).

Each RFID tag, as well as the RFID reader, contain coils, and the alignment between the coil of the tag and the one inside the reader also influences the accuracy of the reading. Three possible alignments of tag and reader are shown in Figure 12c. For case 1, the tag coil and the reader coils are neatly aligned, and the accuracy of the reading is 100%, in 15 tests. If the tag coil and reader coil are unaligned, the accuracy drops to 74% (case 2—barely aligned) or even 13% (case 3—unaligned coils).

Lastly, the speed of the movement can have an effect on the accuracy of detecting RFID tags. For speeds between 0.4 and 0.8 m/s, the accuracy was 100%, provided that the coils were aligned, as shown in Figure 12c, case 1.

To activate/deactivate *Inside mode*, the user says “Inside mode enable” or “Inside mode disable”.

### 3.4. Emergency Alert

The *Emergency alert* functionality is available only when the smart assistant is connected to the Internet. It has two working scenarios: *Alert* and *Notification*. When the assistant receives the “Alert” voice command from the user, it transmits an emergency email to authorized personnel, such as their physician or to emergency services (police, fire department).

The *Notification* scenario is activated through the vocal message “Send a message to …”. Any contact from the contact list is a possible recipient of the email message. An email is sent to the nominated contact, informing them that the user wishes to have a conversation in the near future.

### 3.5. Weather Info

When the user wants to have an outdoor walk, the *Weather info* functionality provides weather related news and updates. Activation of this functionality starts with the vocal command: “Weather, please”. The user is informed about atmospheric conditions, such as outside temperature, humidity, rain appearance, etc. Weather info doesn’t work offline. If the Internet connection is poor or unavailable, the smart cane user will hear the message “I can’t receive data from server”.

### 3.6. Market Mode

With the help of simple vocal commands like “Add 3 kilos of banana to the list”, the user can create a shopping list. The list can be sent by the smart assistant through the message “Send the market list”, to a friend or to a company specialized in food delivery. The entire list can also be used when groceries are purchased directly, as opposed to being ordered.

To add groceries on the list, the user utters a voice message such as “Add 2 L of milk to the list”. The quantity is a float positive number, the measurement unit can be kilos, liters, pieces; the product can be one of seven that are so far included in the smart assistant: tomato, banana, apple, mango, milk, eggs, and sugar. More products can be added, based on the user’s needs and preferences.

To eliminate the last added position, the user simply says “Delete last element”.

### 3.7. Physical Condition

The *Physical condition* functionality was designed to inform people about their physical condition. With the "Health please" or "Physical condition please" vocal commands, the user is given information about the walked distance, the number of steps and the number of burned kilocalories.

To compute these values, Equation (6) is used, for a 1.8 m tall user.
(6)1 step =82 cm,1320 steps =1 km,1 Kcal =48 steps.

### 3.8. Medication Schedule

*Medication schedule* is designed to help visually impaired people with memory problems. *Continuous alert* and *Event info* are parts of this functionality.

*Continuous alert* has three predefined hours at which the user is notified, through some sound alerts, to take their medication: 9.00, 14.00 and 20.00. The sound alerts remain active until the user confirms taking the medication.

*Event info* informs the user regarding the medication that is scheduled for different moments of the day. Using a vocal command as “List of drugs for” followed by the moment of the day “morning”, “noon”, or “evening”, the user obtains the medication list associated with that particular moment, or no medication, if there is none to be taken.

### 3.9. Bus Station Alert

*Bus station alert* is a helpful functionality when the user travels by bus, for long or short distances. When the user gets on the bus, they need to inform the smart assistant about the bus station they plan to get off at. The bus station can be any point from the personalized database. Using the GPS module, the location of the user is updated in real time. When the distance between the bus station and the user is less than 300 m, the smart assistant informs the user to be prepared to get off the bus.

The message delivered to the user is “In X meters is the bus station Y”, where X is less than 300 and Y is the bus station set as destination.

### 3.10. Time and Date

*Time and date* is the simplest functionality of the smart assistant. Using vocal commands as “Time now” and “Date now”, the user is informed about current hour and current date.

All functionalities of the smart assistant were implemented in order to achieve the proposed goals. Testing the system in real conditions proved that the prototype helps the person in outdoor and indoor environments (day and night), detects obstacles and suggests alternative routes, provides directions in a new environment, assists the person in shopping activities, informs authorized personnel regarding medical status, does not require IT skills and proficiency, and is available to the user non-stop.

## 4. Discussion and Conclusions

Advances in technology can significantly improve the life quality of visually impaired persons. Modern navigation and orientation techniques, along with obstacle recognition and avoidance means are nowadays integrated into small, portable devices that become a real help in daily activities.

The navigation functionalities of the proposed smart assistant were tested in indoor and outdoor controlled environments, under various conditions. To prevent possible incidents or injuries in this prototype phase, two volunteers, blindfolded people, tested all functionalities of the system. Their feedback was collected by means of a survey, as follows:How do you evaluate the communication with the smart assistant? (1 to 5)How do you evaluate the quality of the vocal messages? (1 to 5)How do you evaluate the Weather/Alert/Market/Inside/Obstacle Mode functionality? (1 to 5)Do you think the smart assistant is helpful for the visually impaired? (1 to 5)Does the smart assistant increase the user’s independence in daily activities and tasks? (1 to 5)Does using the smart assistant improve the overall mental state? (1 to 5)

The average value of the answers was 4.25 points, showing that the smart assistant can be a solution that provides significant help to visually impaired people, even if there is still room for improvement, in some areas.

The survey included a final question regarding the price people would be willing to pay for such a system, with three options: under $500, between $500 and $1000, and over $1000. Respondents chose the first two options. The analysis of similar smart assistant systems in [30] reveals that the minimum price for a cane equipped with sensors is $200, while a complete assistive system can cost as much as $6000. The cheapest solution, which provides obstacle avoidance, indoor and outdoor navigation and object recognition is priced at $400.

For the proposed smart assistant, the total cost of the materials is less than $200; $144 for the components in the central unit (Raspberry Pi 3B, GPS module, electronic compass, ESP8266 module, audio card, microphone, speaker, and printed PCB) and $51 for the smart cane (Wemos D1 Mini, battery, distance sensors, RFID reader, reflexive sensor, 3D printed case, and wheel). While the final price may be higher, due to mass production and distribution costs, the proposed system is still more affordable than similar solutions, and provides more functionalities.

Tests showed that, in order to design an accessible environment for the visually impaired, RFID tags should be placed above the floorboards, so that the distance between RFID tags and the reader is as small as possible. When the tags are below the floorboards, a distance greater than 30 mm results in poor detection (40% or lower accuracy). To ensure accurate readings, it is recommended that the diameter of the coil on the RFID tags matches that of the RFID reader.

When moving on a predefined pathway containing obstacles that were either fixed (typical urban objects) or moving (other pedestrians), 95% of obstacles were correctly identified by the lateral sensors of the smart cane. The remaining 5% were those positions in front of the cane, and they are detected by moving the cane, as if it were a regular one.

Since the smart cane is equipped with ultrasonic sensors on each side, obstacles are identified without needing to sweep the cane. The user is thus able to move faster, indoors and out. Considering the length of the smart cane (1.5 m) and the maximum distance from which an obstacle can be detected (40 cm), the total distance between user and obstacle is approximately 2 m, ensuring the safety of the user.

The dimensions of the smart assistant render it portable, easily carried inside a purse or backpack, or even as a bracelet. The central unit is supplied from an ordinary power bank, requiring 5 W/h, while the average energy consumption of the smart cane is 250 mW/h, from a battery. The main functionalities are designed to work offline. The interaction between user and smart assistant is achieved via simple vocal commands, making it suitable for any user, regardless of their IT skills and proficiency. It provides highly reliable navigation and guidance and ensures the user’s safety. Moreover, the functionalities regarding physical condition, medication, shopping activities, and weather information facilitate interaction between the user and the environment.

The proposed system shows its limitations when the user comes across a traffic light without sound alerts, or on uneven roads, where potholes that are lateral with respect to the smart cane cannot be detected. The system is not yet able to automatically take into account changes in accessibility, with regular updates of the customized map being required.

Future developments of this prototype may include increasing the processing capabilities, adding a webcam and optimizing its energy consumption so that common images (e.g., traffic lights) are easily recognized, adding more vocal commands, extending the grocery list, and testing the of system by visually impaired people.

The prototype can help visually impaired people to achieve a high level of independence in daily activities, which, in the long term, can have a positive effect on their quality of life.

## Figures and Tables

**Figure 1 sensors-22-04271-f001:**
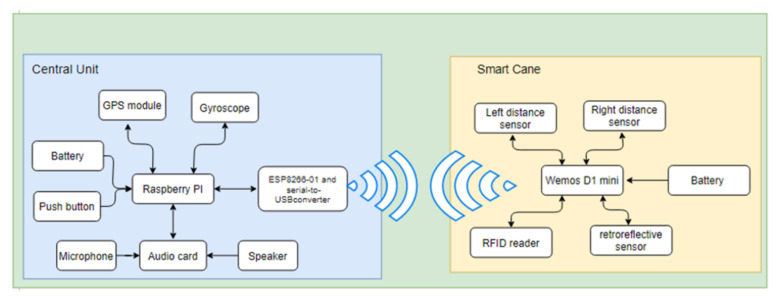
Hardware parts of the smart assistant: the central unit and the smart cane.

**Figure 2 sensors-22-04271-f002:**
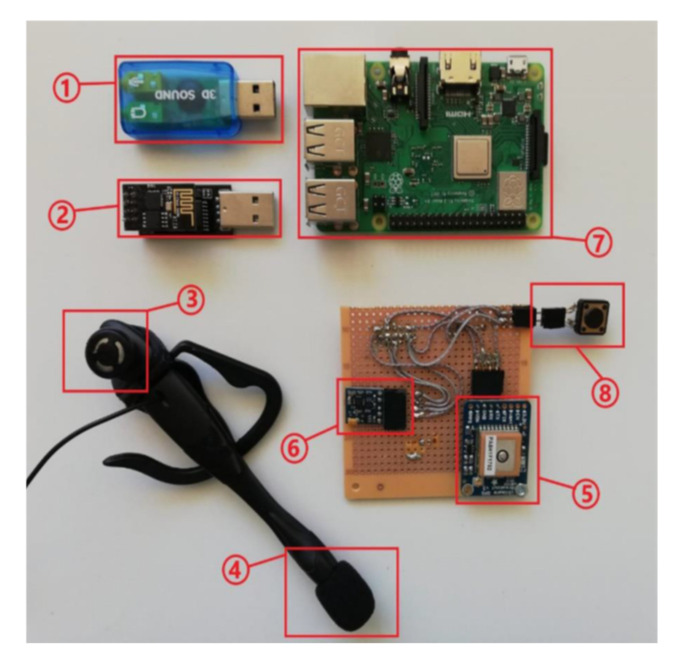
Central unit hardware components: (1) external soundboard with USB interface; (2) ESP8266-01 Wi-Fi module with a serial-to-USB converter; (3) speaker; (4) microphone; (5) GPS Adafruit Ultimate module; (6) electronic compass; (7) Raspberry Pi model 3B+ minicomputer (6), and (8) push button.

**Figure 3 sensors-22-04271-f003:**
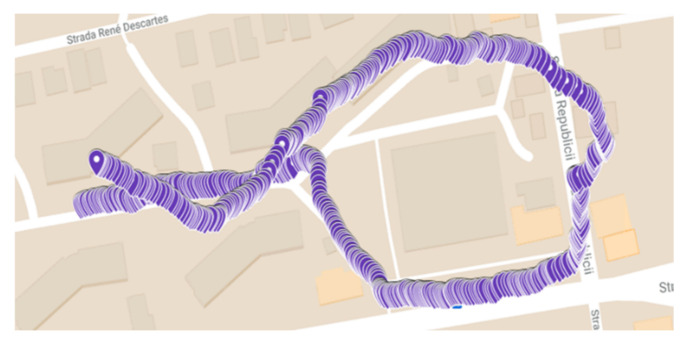
GPS locations provided by Adafruit Ultimate module.

**Figure 4 sensors-22-04271-f004:**
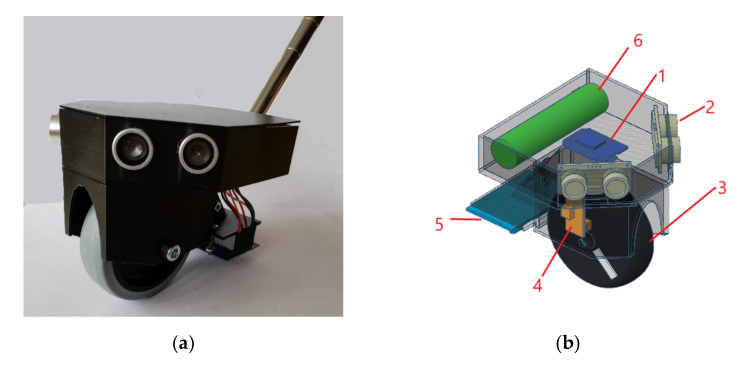
Prototype of the smart cane: (**a**) External view. (**b**) Electronic components: (1) Wemos board, (2) ultrasonic sensor, (3) rubber wheel, (4) optical sensor, (5) RFID reader, (6) battery. (**c**) Circuit diagram: (1) Wemos D1 mini module, (2) HC-SR04 ultrasonic sensors from the left side, (3) HC-SR04 ultrasonic sensors from the right side, (4) MRC522 RFID reader, (5) TCRT5000e reflective optical sensor, (6) 5 V portable power bank.

**Figure 5 sensors-22-04271-f005:**
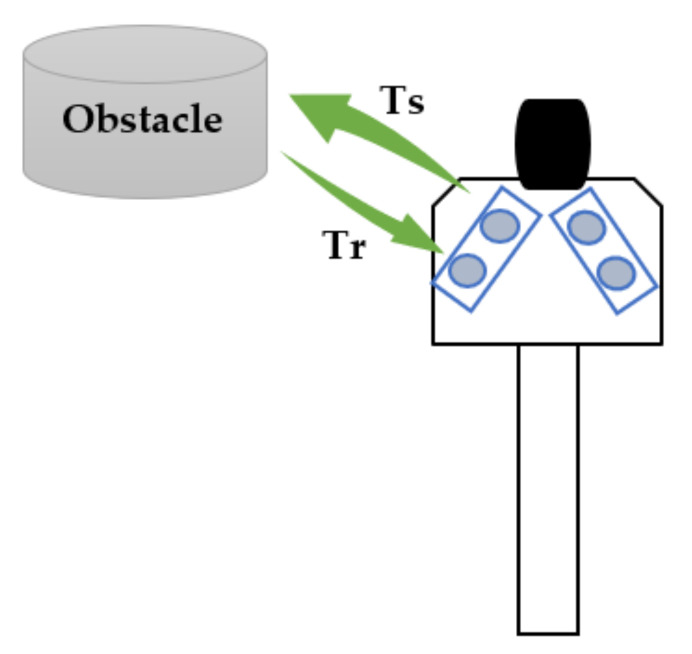
HC-S04 ultrasonic sensor from smart cane—sending and receiving signals: *Ts*—sending time, *Tr*—receiving time.

**Figure 6 sensors-22-04271-f006:**
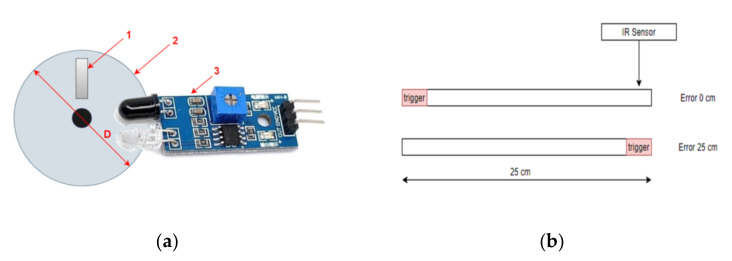
Measuring the distance: (**a**) spinning wheel: (1) trigger, (2) circumference of the wheel, (3) counter; (**b**) minimum error is 0 cm when the trigger is about to be in front of the IR sensor, maximum error is 25 cm when the trigger is directly in front of the IR sensor.

**Figure 7 sensors-22-04271-f007:**
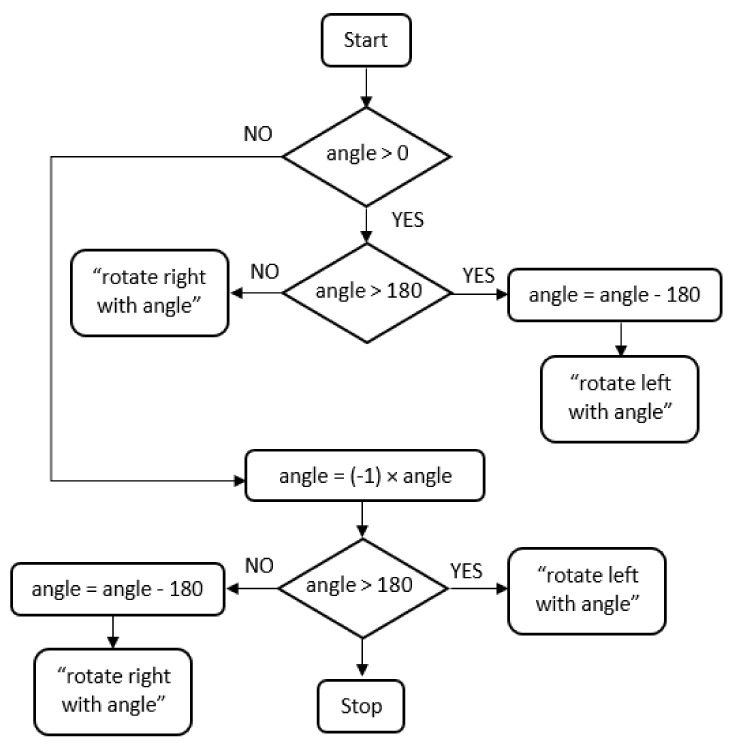
Diagram to set direction.

**Figure 8 sensors-22-04271-f008:**
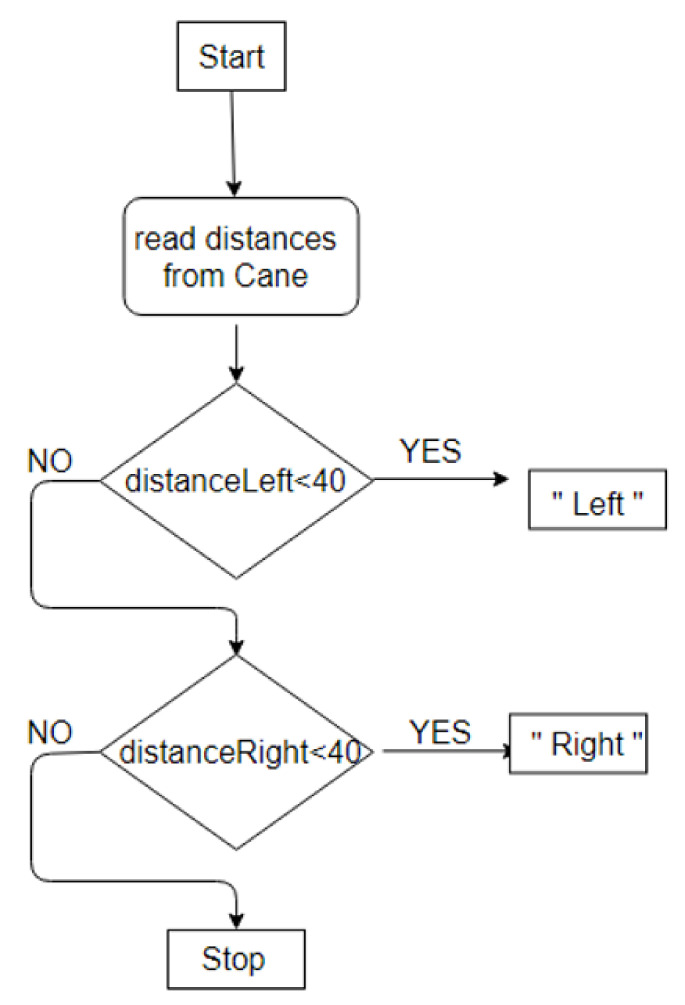
Diagram to detect obstacles.

**Figure 9 sensors-22-04271-f009:**
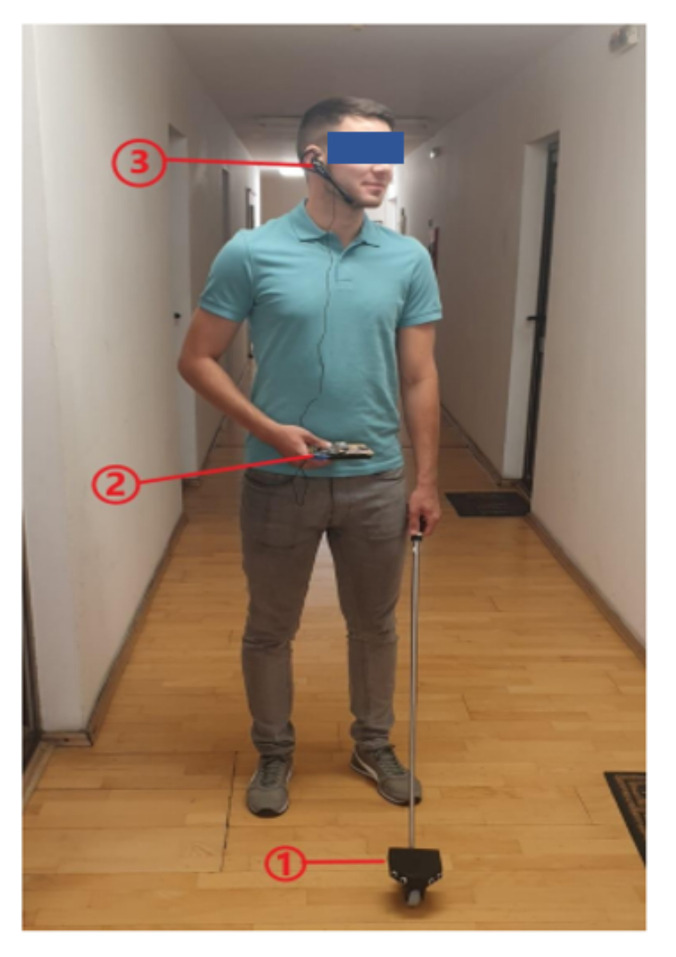
Smart assistant components: (1) smart cane, (2) central unit, (3) headphones and microphone.

**Figure 10 sensors-22-04271-f010:**
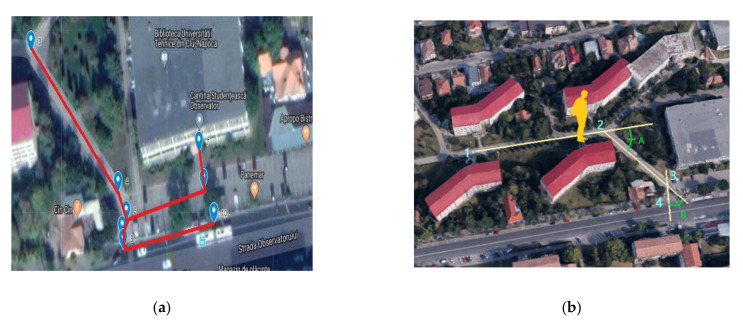
Navigation mode real scenario: (**a**) numbered markers and possible routes using Google My Maps; (**b**) testing *Navigation mode***,** 1—starting point: 2, 3—change direction points: 4—destination point.

**Figure 11 sensors-22-04271-f011:**
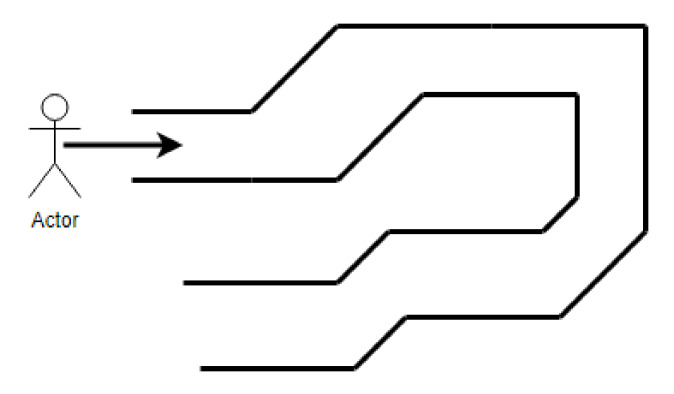
Testing the *Obstacle mode* on a sample pathway, indoor.

**Figure 12 sensors-22-04271-f012:**
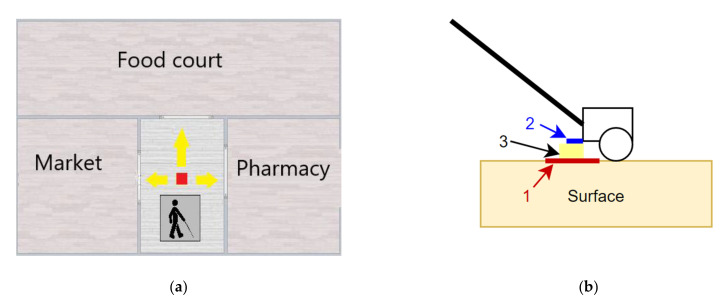
Testing the *Inside mode*: (**a**) the lobby at the entrance of a building; (**b**) testing the accuracy of the reading according to the distance between components - RFID tag (1), RFID reader (2), distance between RFID tag and reader (3); (**c**) testing the accuracy of the reading according to the alignment between the coils of the RFID tag and reader—centers of the coils are perfectly aligned (Case 1), centre of the tag coil barely intersects the coil of the reader (Case 2), unaligned coils (Case 3).

**Table 1 sensors-22-04271-t001:** Summary of components and features of representative electronic travel assistants for visually impaired persons.

Reference	Inertial	GPS	Ultrasonic	Radar	Cam/Phone	RFID	Cane	Indoor	Outdoor	Day, Night	Portable	Alert
Proposed	√	√	√			√	√	√	√	√	√	√
Anandan [7]		√	√		√		√		√			
Saaid [9]						√	√	√	√	√		
Mahida [10]	√				√			√				
Guerrero [11]					√		√					
Kajiwara [13]	√				√	√		√	√			
Tapu [14]					√				√			
Bouteraa [15]			√					√	√	√	√	
Saquib [16]		√	√			√	√	√	√	√	√	
Messaoudi [17]	√		√		√		√	√			√	
Sahoo [21]		√	√				√		√	√		√
Rizvi [22]		√	√					√	√	√	√	√
Castillo [23]		√	√				√					√
Maulana [24]		√	√					√	√	√	√	√
Shahrizan [25]			√		√		√				√	√
Kumar [26]		√	√		√			√	√		√	√
Romadhon [27]		√	√				√		√	√		√
Uddin [28]			√		√			√				
Kammoun [29]		√			√				√		√	
Slade [30]	√	√		√	√		√	√	√	√	√	
Jardak [31]				√				√	√	√	√	
Long [32]				√	√			√	√	√	√	
Cardillo [33]				√			√	√	√	√	√	

**Table 2 sensors-22-04271-t002:** Smart assistant functionalities, commands to activate/deactivate and options.

Functionalities	Commands	Options
Inside mode	Inside mode	Enable
		Disable
Obstacle mode	Obstacle mode	Enable
		Disable
Navigation mode	Set destination to …	Marker 1, 2, …, *n*
	Stop	
Market list mode	Add (1…100) … to the list	Item 1, 2, …, 7…
	Delete last element	
	Send the list	
	Read the list	
Alert mode	Alert	
	Send email	
	Send emergency email	
Time	Time please	
Physical condition	Physical condition please	
Medication	List of medication for … please	Morning
		Noon
		Evening
Weather	Weather please	

**Table 3 sensors-22-04271-t003:** Test cases for voice message recognition.

No.	Noise Level [dB]	Test 1	Test 2	Test 3	Test 4	Test 5	Accuracy [%]
1	20	1	1	1	1	1	100
2	40	1	1	1	1	1	100
3	50	1	1	1	1	1	100
4	60	1	0	1	1	1	80
5	70	1	1	0	1	1	80
6	>75	1	1	0	0	1	60

**Table 4 sensors-22-04271-t004:** Test cases for distance between RFID tag and reader.

Case No.	Tag to Reader Distance [mm]	Accuracy
1	20	99%
2	26	96%
3	31	90%
4	33	40%
5	36	2%

## Data Availability

Not applicable.

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
