# Peer review of "Sensor-Based Prototype of a Smart Assistant for Visually Impaired People—Preliminary Results"

_sensors, 2022, doi:10.3390/s22114271_

Round 1
Reviewer 1 Report
This contribution is focused on a very interesting topic. However, in the scientific literature there are a lot of similar contributions thus the authors should highlight what are the novel elements of their work.
In detail, electronic caned for blind and visually impaired people have been already investigated and proposed by exploiting different technologies. This is one of the last examples reported in the literature:
- Cardillo, C. Li and A. Caddemi, "Millimeter-Wave Radar Cane: A Blind People Aid With Moving Human Recognition Capabilities," in IEEE Journal of Electromagnetics, RF and Microwaves in Medicine and Biology, doi: 10.1109/JERM.2021.3117129.
However, there are a lot of examples concerning other electronic travel aids to assist visually impaired and blind people in autonomous walking, based on the electromagnetic technology (radar), cameras, infrared and ultrasonic sensors. Electromagnetic/radar solutions are completely missing in table 1.
I strongly suggest to perform an accurate comparison with the existing solutions, highlighting pros and cons.
Table 1 should be re-formatted to improve readability.
What is the real measured update rate of the GPS sensor? Although your chips can work up to 5/10 Hz update rate, often update rate higher than 1 Hz are difficult to be achieved.
I think that the well-known messages received from the GPS modules are not relevant in this context.
6.7 m of accuracy seems too high for the safe navigation of a person.
The system seems quite bulky. What is the total weight of the system?
Some discussion concerning the energy consumption of the proposed hardware solutions might be interesting for the readers.
An accurate comparison with the existing solutions within the scientific literature should be provided to highlight the novelty of the contribution.
Reviewer 2 Report
Please see the attached file for comments.

Reviewer 3 Report
The manuscript entitled “Sensor-based prototype of a smart assistant for visually impaired people - preliminary results” developed a smart assistant consisting of an intelligent cane and a central unit; communication between the user and the assistant is carried out through voice messages, making the system suitable for any user. This is written well; however, some issues need to be resolved-
- The technical novelty is not apparent. Please describe the main contribution in detail. The technical part needs more explanation, such as the detection system and algorithms. A small comparison is appreciated.
- The abstract is incomplete. Write down your contribution and novelty in brief. The abstract must be concise and overview of the whole work.
- Correct all references. Cite references according to the sensors template.
- This is a potential work and much needed to help the community. I encourage the author for this exciting solution. I suggest completely preparing the result, submitting the final version, and removing the “preliminary results” from the title.
- Additionally, the readability of the whole paper can be improved.
Round 2
Reviewer 1 Report
The authors addressed all my concerns.
Author Response
Thank you for your feedback.
Reviewer 2 Report
The novelty of this work is insignificant. In this revision, some concerns have still not been addressed.
1. For Eq. (2), the explanations in your responding letter are okay. But, your more clear explanations are not updated in your manuscript. The descriptions for Tr and Ts are confusing in your manuscript.
2. The reviewer agreed that the authors have done a lot of work on this prototype. However, the contributions are limited.
3. The display style for Eqs. (4) and (5) are poor. The reviewer thinks it is not allowed even in a technical report.
4. In Sec. 2.6.1, how do you implement the Dijkstra algorithm to find the shortest path? Based on the map information from Google, the shortest path may not suitable for a blind person. How do you deal with this problem?
Based on my concerns, the reviewer would suggest rejecting this manuscript in its current form.
Reviewer 3 Report
The author carefully revised the manuscript, and I recommend accepting this paper. I hope this work will significantly contribute to the research community.
Author Response
Thank you for your feedback. The authors also hope their work will have a significant impact.
Round 3
Reviewer 2 Report
Thank you for your response. According to the responses, "All nodes are the ones from the customized map, however, computing the shortest distance is not able to take into account accessibility limitations." Then, why do you want to propose the Navigation mode?
